# The Effect of the Terminal Functional Groups on Fluoropolymer on Electrowetting Device Performance

**Eri Oishi [1], Noritoshi Araki [2], Teruya Goto [1], Hiroshi Awano [1] and Tatsuhiro Takahashi [1,*]**

[1]  Department of Organic Materials Science, Graduate School of Organic Materials Science, Yamagata University, 4-3-16 Jonan, Yonezawa, Yamagata 992-8510, Japan
[2]  ARM Technologies Co., Ltd., #213 Sagamihara Incubation Center SIC-1, 5-4-21 Nishihashimoto, Midori-ku, Sagamihara, Kanagawa 252-0131, Japan
*  Correspondence: effort@yz.yamagata-u.ac.jp; Tel.: +81-236-26-3585

**Abstract:** Electrowetting on dielectric (EWOD) devices were fabricated using two hydrophobic organic fluoropolymers, comprising CYTOP (a product name) having different chemical structures only at the terminal functional groups. These devices were subsequently characterized by applying a range of direct current (DC) voltages. The data demonstrated that the EWOD performance was dramatically improved upon incorporating a CYTOP polymer having highly polar terminal functional groups, as compared to a polymer having terminal groups with lower polarity. The new finding about the positive effect of highly polar terminal functional groups on the enhancement of EWOD was exhibited through various careful experiments, changing only the quantitative amount of polar terminal functional groups while keeping other factors constant (thickness, substrate, etc.).

**Keywords:** electrowetting; electrical wettability; fluoropolymer; dielectric polymer; terminal functional groups

## 1. Introduction

Electrowetting on dielectric (EWOD) refers to the phenomenon in which the wetting properties of a liquid droplet (typically composed of water) on a dielectric are modified by an applied voltage. In principle, the dielectric delays electrolysis, allowing for the reversible manipulation of surface wettability. This electrowetting principle can be applied to microfluidic devices [1], electronic paper [2], and energy harvesting [3]. EWOD devices generally consist of a solid electrode surface coated with a thin film of a dielectric inorganic material on which the liquid droplet is placed. However, the majority of inorganic materials having high dielectric constants do not possess good hydrophobic properties, and so a thin hydrophobic organic polymer layer is often applied overtop of the dielectric inorganic film. The change in the contact angle induced by applying a voltage can be estimated using the Young–Lippmann equation,

$$cos\theta = cos\theta_0 + (\varepsilon_0\varepsilon/2t\gamma)V^2, \tag{1}$$

where $\theta$ is the apparent contact angle of the liquid under the applied voltage, $\theta_0$ is the equilibrium contact angle, $\varepsilon_0$ is the permittivity of a vacuum, $\varepsilon$ is the permittivity of the dielectric material, $t$ is the thickness of the dielectric layer, $\gamma$ is the liquid surface tension, and $V$ is the applied voltage.

Recently, the fabrication of electrowetting devices driven by low voltages has received considerable attention. There have been many reports regarding the use of inorganic materials having high dielectric constants, but substances such as barium strontium titanate (BST) [4] and bismuth zinc niobate

(BZN) [5] require complicated synthetic procedures and high annealing temperatures. There have also been several reports regarding the application of the ferroelectric polymer polyvinylidene difluoride (PVDF) [6,7], which can be processed at lower temperatures but tends to exhibit significant hysteresis because of remanent polarization. Other approaches to the development of low voltage electrowetting devices involve the use of droplets incorporating Cd/Se quantum dots [8], although these techniques require careful adjustment of the droplet pH and electrode polarity, which can be complicated. Mibus et al. [9] and Li et al. [10] studied metal oxide/fluoropolymer bilayer EWOD systems based on aluminum oxide and tantalum pentoxide. These devices were fabricated by a simple low temperature anodization process. Using this technique, the thicknesses of the oxide and fluoropolymer layers were readily controlled by varying the anodic time span and the spin-coating conditions, respectively. However, to date, while the effects of the layer thicknesses and of the oxides on EWOD performance have been addressed, the effect of the fluoropolymer itself is not well understood.

In the present work, a low voltage (<30 V) EWOD system consisting of a CYTOP (the product name) layer and an anodic aluminum oxide layer was employed to focus on the effects of the fluoropolymer, because there has been little research regarding hydrophobic organic materials. Two types of CYTOP were used, having identical main chain structures, molecular weights, and molecular weight distributions. The sole difference between the two was the terminal functional groups had different polarities. To ensure that only the effect of the polarity of the terminal groups was examined, all other factors (such as the aluminum oxide preparation process and thickness and the CYTOP layer thickness) were carefully held constant.

We have approached this study to examine the effect of the polar terminal groups on EWOD by only changing the amount of polar functional groups in four experimental ways. The first is comparing two different types of CYTOP, low and high polar terminal groups. The second is comparing CYTOP having low polar terminal groups with and without the initial coating of polar low molecules (amino type silane coupling agent) which contribute to the stronger bonding strength. The third is comparing among two different CYTOPs, those of a single layer and CYTOPs of a double thin layer structure, but keeping total thickness the same. The fourth is a comparison among CYTOPs of thin layer using a solution blend of two types of CYTOPs. The aim of these four approaches was to investigate the effect of polar terminal functional groups of the fluoropolymer on the EWOD device performance with keeping all other factors constant.

## 2. Materials and Methods

### 2.1. Fabrication Method for Basic EWOD Devices

The structure of the EWOD devices is shown in Figure 1, while Figure 2 summarizes the structures and molecular weight of the CYTOP fluoropolymers (Asahi Glass Co., Tokyo, Japan). The terminal functional moieties of the polymer abbreviated CYTOP-A consisted of carboxyl groups (low polar terminal functional group), while the CYTOP-M had terminal amide-silane groups (high polar terminal functional group). Both had the same broad molecular weight range of 150,000 to 200,000 g/mol. Anodization of the aluminum plates (Nilaco Corp., Tokyo, Japan, 100 μm thick) was performed in an aqueous 3 wt % tartaric acid solution for 1 h at 30 V at room temperature, to form thin films approximately 48 nm thick. All substrates were carefully prepared with the same condition to keep surrounding factors constant. Each CYTOP was dissolved in CT-Solv 180 (Asahi Glass Co., Tokyo, Japan) to a concentration of 3 wt % and then spin-coated onto the substrates at 3000 rpm for 20 s, followed by heating for 1 h at 80 °C and curing for 1 h at 180 °C to form films approximately 120 nm thick. The thickness of each film was carefully measured using a Dektak X stylus surface profiler (BRUKER).

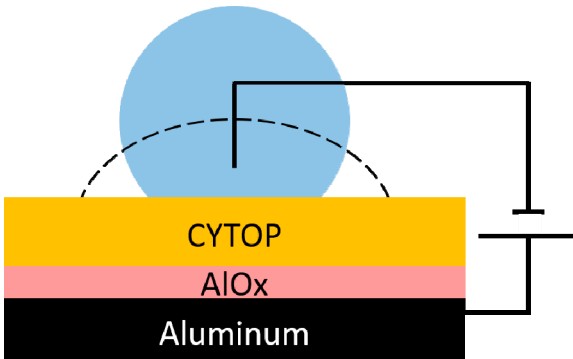

**Figure 1.** A schematic of the electrowetting on dielectric (EWOD) device used to assess changes in contact angle upon applying a voltage.

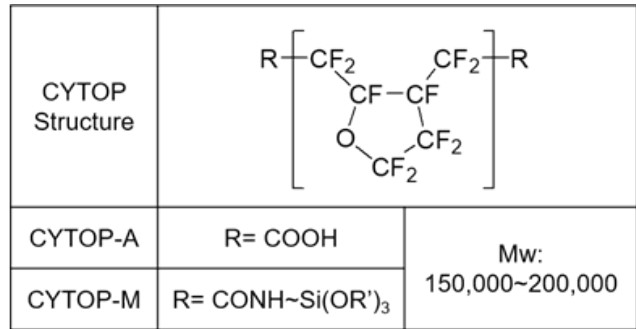

**Figure 2.** The structure and molecular weight of the CYTOP fluoropolymer. R indicates the terminal functional groups, which (as shown) were carboxyl groups in CYTOP-A (low polar terminal functional group) and amide groups in CYTOP-M (high polar terminal functional group).

### 2.2. Silane Coupling Treatment for Formulation of an Amide Bond to the CYTOP-A/AlO$_x$ Interface

For the examination of effect of the bonding strength and polar groups using low molecules at the CYTOP/AlO$_x$ interface on EWOD, 1 wt % 3-aminopropyltriethoxysilane (Tokyo Chemical Industry Co., Ltd, Tokyo, Japan)/ ethanol solution was spin-coated onto AlO$_x$ at 3000 rpm for 20 s after anodization of Al plates. Then, CYTOP-A was spin-coated onto aminosilane/AlO$_x$ at 3000 rpm for 20 s, and heating for 1 h at 80 °C and curing for 1 h at 180 °C.

### 2.3. Double Layer Formation Using Two Types of CYTOP

After Al anodization, 0.5 wt % CYTOP-A solution was spin-coated at 1000 rpm for 20 s, and heating for 30 min at 50 °C. A 3 wt % CYTOP-M solution was also spin-coated onto coated CYTOP-A at 2000 rpm for 20s, and heating for 1 h at 80 °C and curing for 1 h at 180 °C. In this way, a double layer structure of the CYTOP-M layer and CYTOP-A layer was fabricated keeping the total thickness constant (120 nm). The device has carboxyl groups at the CYTOP/AlO$_x$ interface, and amide groups inside a CYTOP films.

### 2.4. Thin Layer Preparation Using a Solution Blended with Two Types of CYTOP

In this section, in order to correlate amide group (high polar) and carboxyl group (low polar) quantities, CYTOP-M and CYTOP-A were mixed in a solution at the ratio of M/A is 1/3, 1/1, and 3/1. Each CYTOP was solved in CT-Solv. 180 at 3 wt %. All blended CYTOP solutions were spin-coated onto AlO$_x$ at 3000 rpm for 20 s, heated for 1 h at 80 °C, and then cured for 1 h at 180 °C.

*2.5. Characterization and Electrowetting Test*

The contact angles on the various devices were determined with a Theta Lite contact angle meter (Biolin Scientific, Gothenburg, Sweden). Electrowetting measurements were performed in air by placing a 3 μL droplet of water onto a CYTOP/AlO$_x$/Al stack and applying a DC voltage between the Al substrate and an inset Pt probe in contact with the water droplet. In these trials, the voltage was increased in 5 V increments. Photographic images of the contact angle were acquired at each applied DC voltage over a span of 10 s, while increasing the voltage stepwise throughout the entire range of stable operation. The dielectric constants of the CYTOP specimens were also ascertained at 10 kHz using an LCR meter (NF Co., Kanagawa, Japan).

## 3. Results and Discussion

In initial work, the thickness of the CYTOP layer was optimized. Overly thin films can sometime result in pinhole defects due to the surface roughness of the aluminum plate, leading to breakdown of the device, while overly thick films require the application of a high voltage. Taking these two factors into consideration, an optimal thickness of 120 nm thick was selected (the thin layer does not have pinholes and defects, measured by AFM), so as to avoid pinholes while allowing the use of a low driving voltage. The thickness of each sample was carefully tuned by adjusting the spin coating conditions.

*3.1. Comparison between CYTOP-M and CYTOP-A*

Specimens were made by placing either CYTOP-A or CYTOP-M thin layers (having equivalent thicknesses) on top of AlO$_x$ substrates. Figure 3a shows the results of contact angle measurements at various applied DC voltages. In the case of the CYTOP-A layer, the film thickness was 116 nm and the initial contact angle (at 0 V, Figure 3a, (ii)) was 108.9° but changed to 95.4° in conjunction with the maximum wetting state at a voltage of 40 V (Figure 3a, (iv)). This specimen exhibited breakdown behavior at 45 V, as determined by the visual observation of gas generation in the water droplet. The overall change in the contact angle on going from 0 to 40 V was only 13.5°, suggesting relatively low EWOD performance. This small variation in the contact angle was quite unexpected, since all prior studies using CYTOP have shown large contact angle changes. We therefore repeated the experiment several times to confirm reproducibility. CYTOP is generally considered to be a good hydrophobic polymer for EWOD applications, although it appears that CYTOP-A may not have been used in EWOD devices in previous works.

The same trials were carefully performed using the CYTOP-M version. A film having a thickness of 118 nm showed an initial contact angle of 110.6° (Figure 3a, (i)) and this value changed to 72.3° at an applied voltage of 25 V (Figure 3a, (iii)), equivalent to a change of 38.3°. Again, the reproducibility of these results was confirmed. Thus, unexpectedly, the data show that there was a significant difference in performance between CYTOP-A and CYTOP-M, such that the CYTOP-M sample exhibited much better EWOD performance (i.e., a larger contact angle change). In addition, the breakdown voltage of the CYTOP-M unit was 30 V, which was lower than that obtained using the CYTOP-A (45 V).

It should be noted that variations in the AlO$_x$ substrate surfaces, which were generated in our laboratory via the anodization of Al plates, could also have affected the EWOD performance. However, each of the Al plates used in these trials underwent anodization at a low voltage of 30 V for exactly 1 h. The anodized Als thicknesses were 48 ± 2 nm. Thus, it is reasonable to conclude that the significant difference in the polarities of the terminal functional groups (that is, the amide-silane and carboxyl moieties) were primarily responsible for the large difference in the contact angle change. Specifically, the presence of the more polar amide-silane end groups resulted in a lower breakdown voltage.

Here, it is helpful to consider Equation (1), provided above. Despite the large difference in the contact angle change, the dielectric constants of both the CYTOP-M and CYTOP-A were found to be approximately 2.1 based on measurements using an LCR meter over a wide frequency range (100 kHz to 10 Hz; Figure 4). The solid curve in Figure 3a shows the theoretical curve obtained using the Young–Lippmann equation for both samples, and indicates that the results obtained from the EWOD device using CYTOP-A cannot be explained by the equation. It should be noted that Equation (1) does not take the effect of the $AlO_x$ layer into account, although we believe that this is acceptable since, in the case of a very thin $AlO_x$ layer, the oxide has a negligible effect on performance below a specific threshold voltage [9]. In addition, as noted above, each $AlO_x$ film should have been identical and so any effect of the oxide would have been the same for each sample. For these reasons, we consider it reasonable to construct the ideal curve shown in the figure based solely on variations in the CYTOP type, layer thickness, and dielectric constant. Interestingly, the ideal curve from Equation (1) fits the performance obtained using the CYTOP-M very well.

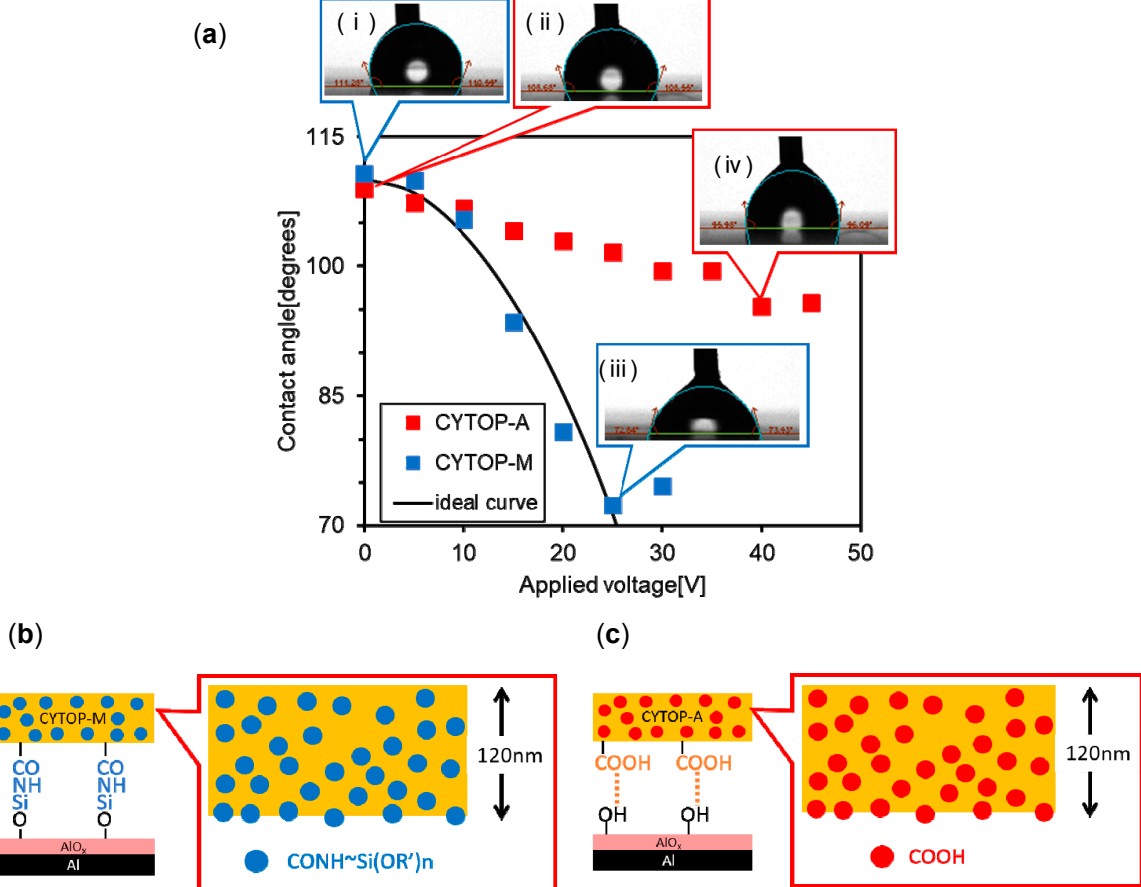

**Figure 3.** (**a**) Contact angles as functions of applied DC voltage for both CYTOP-A (red) (low polar terminal group) and CYTOP-M (blue) (high polar terminal group), increasing the voltage in 5 V increments. The black line indicates the ideal theoretical curve as calculated using the Young–Lippmann equation. (i) Initial contact angle image of CYTOP-M. (ii) Initial contact angel image of CYTOP-A. (iii) Contact angle image of CYTOP-M at applied 25 V. (iv) Contact angle image of CYTOP-A at applied 40 V. (**b**) Stacked CYTOP-M/$AlO_x$/Al sample model, the interface of CYTOP-M/$AlO_x$ was covalently bonded. Blue dots show high polar functional terminal groups. (**c**) Stacked CYTOP-A/$AlO_x$/Al sample model, the interface of CYTOP-A/$AlO_x$ was held together by hydrogen bonding. Red dots show low polar functional terminal groups.

The presence of less polar terminal functional groups in the CYTOP would be expected to have a minimal effect, since these groups account for a relatively small percentage of the overall mass of the polymeric layer. Despite this, CYTOP-M and CYTOP-A gave very different results even though they had approximately the same thickness and dielectric constant values. In previous works, the electrowetting curves obtained using CYTOP have matched the ideal curve, suggesting that the CYTOP specimens employed in these prior studies were all of the CYTOP-M type. Therefore, it appears that even a small proportion of polar functional groups in the thin layer can result in a large contact angle change (meaning improved EWOD performance).

Hence, focusing on the interface between each CYTOP and $AlO_x$, the bonding state was different (Figure 3b,c). The CYTOP-M/$AlO_x$ interface has covalent bonding, on the other hand, the CYTOP-A/$AlO_x$ interface has hydrogen bonding. So, the two samples' properties were different not only in terminal functional groups but also the bonding state. In the following section, we tried additional experiments about effects of bonding state and kinds of terminal functional groups.

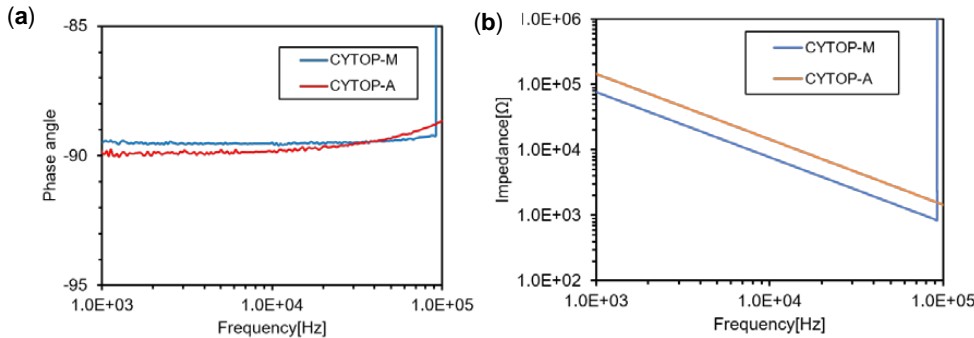

**Figure 4.** (**a**) Phase angle–frequency curve. (**b**) Impedance–frequency curve. The dielectric constant of CYTOP-M and CYTOP-A were calculated by (**a**) and (**b**) curves at 10 kHz, both CYTOP dielectric were almost same, about 2.1.

### 3.2. Effects of Amino Silane Coupling Treatment on the CYTOP-A/$AlO_x$ Interface

Figure 5a shows the results of contact angle measurements at various applied DC voltages, where blue dots are CYTOP-M/$AlO_x$, red dots are CYTOP-A/$AlO_x$, and green dots are CYTOP-A/amino silane (AS)/$AlO_x$. The AS layer was too thin to measure, but the total thickness of the AS layer and CYTOP-A layer was 120 nm. The CYTOP-A/AS/$AlO_x$ sample showed an initial contact angle of 108.1° and this value changed to 75.5° at an applied voltage of 30 V, equivalent to a change of 32.6°. So, EWOD performance with CYTOP-A was improved by amino silane coupling treatment, which is low molecules having high polar groups. In addition, the bonding state of the CYTOP/$AlO_x$ interface changed to covalent from hydrogen bonding by AS (Figure 5b). It was clearly suggested that the CYTOP/$AlO_x$ interface with covalent-bonding or amide groups (introduction of high polar groups) was important for high-performance EWOD devices.

### 3.3. Effect of the Double Layer Structure of CYTOP-A and CYTOP-M

Figure 6a shows the results of contact angle measurements at various applied DC voltages, blue dots are CYTOP-M/$AlO_x$, red dots are CYTOP-A/$AlO_x$, and yellow dots are CYTOP-M/CYTOP-A/$AlO_x$. The CYTOP-A thinner layer was 20 nm, and total CYTOP layer thick was 120 nm. CYTOP-M/CYTOP-A/$AlO_x$ sample showed an initial contact angle of 112.7° and this value changed to 81.3° at an applied voltage of 25 V, equivalent to a change of 31.4°. The sample had hydrogen bonding between CYTOP and $AlO_x$, because $AlO_x$ was contacted with CYTOP-A only (Figure 6b). The hypothetical interpretation about the excellent EWOD performance was the strong interface bonding based on the results of Section 3.2. However, the result in this section from the double layer of CYTOP-A at the interface (weak bond) and CYTOP-M at the above layer suggested that the excellent EWOD is induced by the amounts of polar groups in the volume of thin layer, not by the interface bonding.

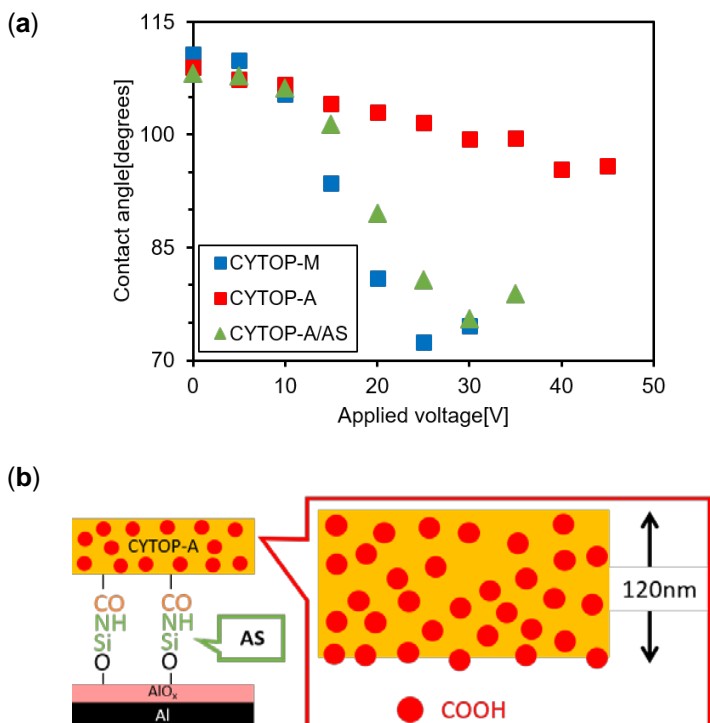

**Figure 5.** (**a**) EWOD curve of CYTOP-M (blue), CYTOP-M (red), and CYTOP-A/AS (green). All samples were placed on AlOₓ/Al. (**b**) Stacked CYTOP-A/AS/AlOₓ/Al device structure model. The interface of CYTOP-A/AlOₓ was covalently bonded by adding AS between CYTOP-A and AlOₓ, and the high polar groups (amino silane) were placed at the interface.

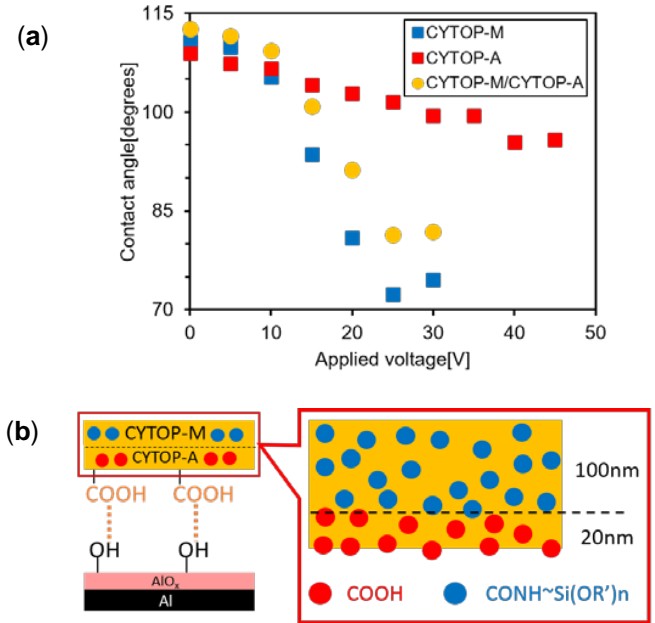

**Figure 6.** (**a**) EWOD curve of CYTOP-M (blue), CYTOP-A (red), and CYTOP-M/CYTOP-A (yellow). (**b**) The model of stacked CYTOP-M/CYTOP-A/AlOₓ/Al. The CYTOP/AlOₓ interface had hydrogen bonding because of lowly placed CYTOP-A with carboxyl groups, and amide groups were included inside a CYTOP layer.

### 3.4. Correlation of Amide Groups (High Polar) and Carboxy Groups (Low Polar) Quantities

To examine the hypothesis that the amounts of total polar groups in thin layer determines EWOD performance, additional experiments were performed. Two types of CYTOP were blended in solution, the ratio of CYTOP-M:CYTOP-A = 3:1, 1:1, and 1:3 (wt %). The results of the contact angle by applying DC voltage is shown in Figure 7a, and Figure 7b shows the M:A = 1:1 version. Then, for an easy to understand correlation of CYTOP-M and CYTOP-A quantities, Figure 8 shows the CYTOP-M amount as the x-axis, and the amount of contact angle reduction as the y-axis. In this case, the EWOD performance was improved relative to that obtained from the original CYTOP-A sample, and the performance was found to be increasingly enhanced as the CYTOP-M proportion in the mixture was increased until 50 wt %. On the other hand, when the CYTOP-M amount exceeds 50 wt %, the contact angle change was almost constant. The results showed that the polymer with amide groups is better dielectric for higher performance EWOD devices.

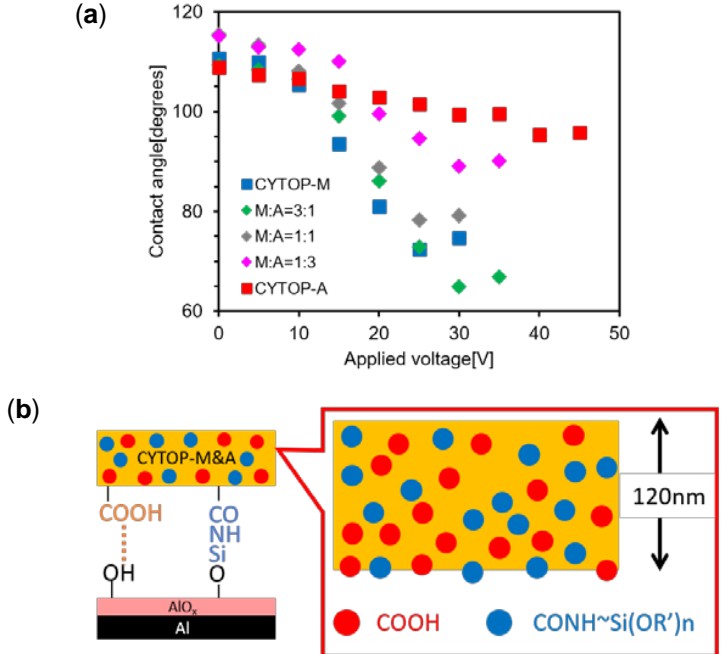

**Figure 7.** (**a**) Results of EWOD curve, with M:A = 3:1 (green), M:A = 1:1 (gray), and M:A = 1:3 (pink). Each sample had breakdown at 30 V or 35 V. All samples had a DC voltage applied, increasing the voltage in 5 V increments. (**b**) The model of the CYTOP ratio was M:A = 1:1. CYTOP-M and CYTOP-A are uniformly dispersed.

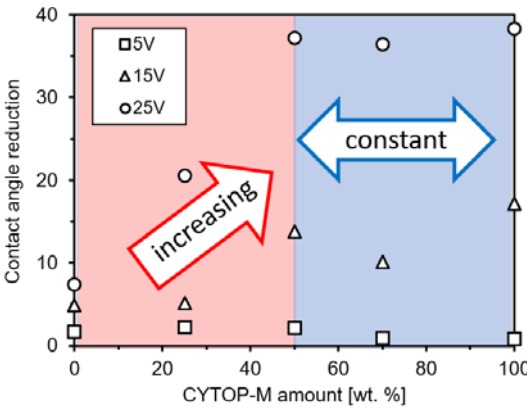

**Figure 8.** Correlation graph between contact angle reduction and CYTOP-M amount. Square dots mean 5 V applied, triangle dots 15 V applied, and circle dots 25 V applied.

## 4. Conclusions

We demonstrated the effect of functional groups on EWOD performance by using CYTOP-M and CYTOP-A, then we fabricated various EWOD devices with a CYTOP mono layer, silane coupling treatment on $AlO_x$, CYTOP double layer, and blended CYTOP. These results clearly demonstrated that the quantitative amounts of high polar groups (amide groups) is the key important factor for the improvement of the EWOD performance, presumably because amid groups are more polar than carboxyl groups. Thus, it is firstly quantitatively demonstrated that the amounts of amide groups in hydrophobic polymer layer governs EWOD performance (the place where amide groups exist does not matter), which is easily controlled by the terminal groups, CYTOP-M, which is better suited to applications in low voltage EWOD devices. We expect that hydrocarbon polymer-modified amide groups can be used as a dielectric polymer for EWOD.

**Author Contributions:** E.O. performed the measurements, analyzed experimental data, and wrote the manuscript; N.A. and T.T. supervised the entire project; and T.G. and H.A. participated in comprehensive discussions, and provided helpful advice and suggestions.

**Funding:** This research received no external funding.

**Acknowledgments:** The authors acknowledge Tomohito Sekine, Yuta Matsushima, and Daisuke Yokoyama of Yamagata University for valuable advice and suggestions regarding each measurement and analyzed experimental data.

**Conflicts of Interest:** The authors declare no conflict of interest.

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
