# Peer review of "The Effect of the Terminal Functional Groups on Fluoropolymer on Electrowetting Device Performance"

_technologies, doi:10.3390/technologies7030052_

Round 1

Reviewer 1 Report

This manuscript describes the fabrication of EWOD devices using two hydrophobic organic fluoropolymers comprising CYTOP having different chemical structures at the terminal functional groups. The devices were characterized by applying a range of DC voltages, which demonstrated that the EWOD performance was improved upon incorporating a CYTOP polymer having highly polar terminal functional groups, as compared to a polymer having terminal groups with lower polarity. I recommend the publication of this manuscript after the following minor revisions.

1.    A formal name of the abbreviation, CYTOP, should be described when it is first appeared in the text.

2.    The beginning part of the abstract is like the introduction, which should be revised as more suitably for the abstract.

3.    The future aspect of the present work should be added at the last part of the conclusion.

Author Response

1.    A formal name of the abbreviation, CYTOP, should be described when it is first appeared in the text.

 →”CYTOP” is a product name of Asahi Glass Co.

2.    The beginning part of the abstract is like the introduction, which should be revised as more suitably for the abstract.

We agree with you that the beginning part of abstract is like the introduction, then we deleted beginning part of the abstract.

3.    The future aspect of the present work should be added at the last part of the conclusion.

We revised the last part of conclusion (line 237)

Reviewer 2 Report

In this article, the electrowetting on dielectric devices fabricated with two hydrophobic organic fluoropolymers having different chemical structures only at the terminal functional groups, 
has been systematically studied. A significant improvement of the electrowetting is observed with the polar terminal functional groups.

The manuscript is well organized and free of significant grammatical errors or misspelling. The results are scientifically and technologically interesting. It could be published after minor corrections:

* It is affirmed that “… double  layer structure of CYTOP-M layer and CYTOP-A layer was fabricated keeping the total thickness 
constant (120nm). The device has carboxyl groups at CYTOP/AlOx interface, and amide groups inside a CYTOP films”. What experimental evidences assure this structure? Could these films have defects and inhomogeneities? 

*Photographic images of the contact angle were acquired at each applied DC voltaje to measure the contact angle. Some illustrative examples of these photographies should be shown.

*Fig. 4(b) is not mentioned in the manuscript.

* Replace the word “voltaje” by “voltage”

Author Response

In this article, the electrowetting on dielectric devices fabricated with two hydrophobic organic fluoropolymers having different chemical structures only at the terminal functional groups, has been systematically studied. A significant improvement of the electrowetting is observed with the polar terminal functional groups.

The manuscript is well organized and free of significant grammatical errors or misspelling. The results are scientifically and technologically interesting. It could be published after minor corrections:

* It is affirmed that “… double  layer structure of CYTOP-M layer and CYTOP-A layer was fabricated keeping the total thickness constant (120nm). The device has carboxyl groups at CYTOP/AlOx interface, and amide groups inside a CYTOP films”. What experimental evidences assure this structure? Could these films have defects and inhomogeneities?

In general, terminal groups of polymer is located in surface of polymer film, the surface means the top of surface or the interface stacked layers. The part of 2.1~2.3 in this paper, we discussed the effect of the interface CYTOP and AlOx. We can explain the evidence of terminal groups is located in the interface, because different terminal groups (-CF3) CYTOP (it is called CYTOP-S) has peel property, therefore CYTOP-S was used as mold lubricant. It shows that terminal groups are related in strength of interfacial adhesion. So we conclude that terminal groups are located in interface of CYTOP/AlOx.

*Photographic images of the contact angle were acquired at each applied DC voltaje to measure the contact angle. Some illustrative examples of these photographies should be shown.

We added some photograph in Fig. 2(a)-()~().

*Fig. 4(b) is not mentioned in the manuscript.

Thank you for your notice. We added it in line 195.

* Replace the word “voltaje” by “voltage”

We revised it.

Reviewer 3 Report

This work compares two kinds of CYTOP with different concentrations of functional group on electrowetting performance. Four experiments were conducted to make sure the performance enhancement coming from the content of functional group, not by other factors such as the thickness or bonding type. The topic and the results should fit the interests in related field with reasonable novelty.  I recommend the acceptance of this manuscript with some double checking on spelling, such as

line 212, (week bond), should be (weak bond). 

Author Response

Thank you for your notice. We revised it

Reviewer 4 Report

This article designs a systematic method to the effect of the terminal functional groups on electrowetting on dielectric (EWOD) devices, specifically CYTOP the fluoropolymer. The results demonstrated that the high polar terminal functional groups, i.e. amide-saline groups, majorly improves the overall EWOD performance, while its counterpart, the low polar terminal functional groups, i.e. carboxyl groups, impacts little. This article focuses on the effect of polarity of the fluoropolymer, which has not been widely noticed, and the results of these experiments satisfactorily fill the gap.

The experiment is composed with four comparative groups with different polarity and has a rigorous academic logic, though a few details could be elaborated further. In Part 3.1, the minor effect of AlOx is neglected with few sentences, which could be a little restricted for readers who peruse. In Part 3.3, the effect of double layer structure of CYTOP-A and CYTOP-M is testified. And result of the layer of CYTOP-M/CYTOP-A leads to the conclusion that the EWOD is induced by the quantity of polar groups in thin layer not the interface bonding. While the mixed layer that the CYTOP-A at the above and CYTOP-M at bottom is not included in the control groups, which might cause unconvinced conclusion drawing. It did include the effect of CYTOP-A/AS/AlOx sample in Part 3.2, however, due to the extremely thin AS layer, it is difficult to form an equal connection with the 20nm CYTOP-A in mixed double layer in Part 3.3.

Overall, this article makes an impression on the effect of functional groups in EWOD performance using CYTOP-A and CYTOP-M. It can be published with minor revision.

Author Response

Thank you so much for your comment.

 From result part 3.1, we expected that EWOD performance was improved by stronger bonding (covalent bonding) at CYTOP/AlOx interface. It showed the same result in part 3.2. However, part 3.3 shows a good EWOD performance despite the CYTOP-M/CYTOP-A/AlOx stacked device has weak bonding (hydrogen bonding) at the CYTOP/AlOx. From the part 3.3 result, we have a possible that EWOD performance was improved by including amide groups in dielectric polymer, so, the place of amide group is no matter where, at the interface CYTOP/AlOx, the top of surface, and inside a layer. To confirm it, in the part 3.4, we blended CYTOP-A and CYTOP-M.

Certainly, it was difficult to understand, sorry. Then, we revised conclusion a little.